# Client preferences in noncommunicable diseases management in Australia: A scoping review

Aklilu Endalamaw [1,2]*, Darsy Darssan[1], Resham B. Khatri[1], Yibeltal Assefa[1]

1 School of Public Health, the University of Queensland, Brisbane, Australia, 2 College of Medicine and Health Sciences, Bahir Dar University, Bahir Dar, Ethiopia

* yaklilu12@gmail.com

## Abstract

Noncommunicable diseases (NCD) are the leading causes of mortality in Australia. Understanding clients' preferences is essential to implement effective care models that revitalize utilization of services. This study aims to review existing evidence on client preferences of service delivery among people with NCD in Australia. We followed PRISMA extension for scoping review. Articles conducted with discrete choice experiment were included. Clients with NCD in general or one of the five major diseases: cardiac diseases, diabetes, cancer, chronic respiratory diseases, and mental health disorders. We used the Differentiated Service Delivery Framework to synthesize the findings. Clients expressed a preference for better-trained health workers to handle sample collection and referrals rather than doing it themselves. For polygenic cancer testing, clients preferred primary care physicians over genetic specialists. There was a preference for a model of care that engaged clients in the decision-making process, safe, comprehensive, effective, and affordable services delivered closer to the community, and exhibited shorter waiting times to receive care. Clients preferred face-to-face presence for anxiety and depression screening, polygenic testing for cancer, and follow-up care for breast cancer survivor. Clients preferred less frequent follow-up appointments except those with NCD that needs close follow-up. Clients need a health system that prioritizes patient-centered and community-based models that enhance accessibility, affordability, and safety. Reducing wait times, offering flexible follow-ups and face-to-face services may improve patient satisfaction, trust, and treatment adherence. Failing to align services with patient preferences may lead to lower engagement and reduce healthcare effectiveness.

## Introduction

Service delivery for noncommunicable diseases (NCD) encompasses multifaceted approaches, including primary healthcare (PHC), specialist consultations, community-based programs, patient education and self-management support [1,2].

**Data availability statement:** All data are in the manuscript and/or supporting information files.

**Funding:** The author(s) received no specific funding for this work.

**Competing interests:** The authors have declared that no competing interests exist.

Countries follow the World Health Organization's (WHO) regional NCD prevention and control framework, 'aimed at turning a sick-system to a health system to end the NCD epidemic' [2]. In line with the WHO's framework, Australia has identified three prioritized objectives for chronic conditions, including activities focusing on health promotion and disease prevention; efficient, effective, and appropriate care; and targeted interventions for priority populations [3]. Australia's health system ranked third highest among eleven high-income countries in 2021, based on performance indicators such as access to care, quality, efficiency, equity and health outcomes [4]. A research published in 2020 revealed that 72% of people with diabetes in Australia were diagnosed, and about 50% received standard care [5]. By providing a better care, Australia estimated to reduce premature mortality due to NCD by 25% by the end of 2025 from the 2010 baseline [6]. In this country, life expectancy at birth was also projected to be increased by 5.9 years in 2019 compared to the 1990 baseline [7].

Nevertheless, NCD remain the leading cause of mortality in Australia, accounting for over 90% of total deaths in 2019 [7]. There have been challenges related to timely screening for individuals living in remote areas and inequities in access. Diabetes screening among high risk group was 55.2% [8]. According to the PHC Advisory Group's report in 2015, individuals with chronic conditions experienced uncoordinated care, difficulty in accessing services due to lack of mobility and transport (remoteness), language, and feelings of disempowerment, frustration and disengagement [9]. Fisher and colleagues further noted Australia's 'episodic primary medical care' as 'a poor model of care for NCD' because of its focuses on biomedically-oriented general practitioner services and hospitals, with a concomitant lack of attention on other issues affecting people with NCD [10].

As a result, the fragmentation in service delivery and the lack of connection between different levels of care recommended to be addressed in the PHC system 10 Year's Plan [11]. In addition, identifying tailored strategies and revitalizing the routine model of care is essential. This can be achieved through the involvement of clients and community members, in which preferences of client with NCD could be identified and ensured [12,13]. Once client's preferences and values are identified, health care professionals can integrate values of adults with NCD in primary care through approaches of concern, competence, communication, and congruence [14]. Incorporating client preferences fosters shared decision-making, guide treatment choices, enhance satisfaction, increase treatment completion, and improve clinical outcomes [15,16].

Hence, we identified clients' preferences for NCD management in Australia. The findings from this review will support in developing a client-centered model of care.

## Methods

### Reporting

We used PRISMA extension for Scoping Reviews (PRISMA-ScR) to report this review (S1 PRISMA Checklist) [17].

## Eligibility criteria

We set eligibility criteria based on the objectives, population, context, and outcome (PCC). Population denotes the study population; context refers to the setting or location of the study; and outcome refers to the outcome variable or the main issue under study [18]. We included preference articles used discrete choice experiment (DCE) among clients with major NCD in general or specifically on cardiovascular diseases (CVD), cancer, chronic obstructive pulmonary diseases (COPD), diabetes, or mental health disorders. Articles of any study design conducted in Australia and published in English before the last search date were included. The most recent search date was on October 23, 2024. Published articles were included regardless of the study population's residence, gender, language spoken, age, religion, occupation, employment, other social status, and severity of clinical conditions. We excluded conference proceedings, abstract and citation only, commentary, editorials, and non-English articles.

## Information sources and search strategies

We searched PubMed, Web of Science, Scopus, and EMBASE articles. We used keywords for the search based on 'discrete choice experiment', and 'NCD'. We did not include Australia' in the search strategy, which was aimed at broadening the number of articles to be screened. Boolean operators, namely AND, OR, quotations, and asterisk were used to build the search strategies. After identifying the appropriate index terms, we adhered to Bramer and colleagues' systematic approach to searching [19]. The search strategies for all databases are presented in the supplementary file (S1 Table).

## Sources of evidence

We exported all available articles from the four databases to the EndNote 20 reference manager [20]. First, duplicates were automatically removed. Then, title and abstracts screening was conducted in line with the eligibility criteria. Authors commented the overall screening process during meetings every week. Finally, the full-text selection was conducted, and data extraction proceeded.

## Data charting and data items

The first author (AE) drafted the data extraction sheet in Microsoft Excel; the last author, YA, commented on it. Then, it was shared with the rest of team members. After the team members approved the data extraction form, AE extracted the characteristics of articles and main findings from the full text of each eligible article. Extracted data were cross-checked to see if there were any discrepancies between extracted data and information from the full-text article. We extracted the first author with publication year, data collection period (year of study), study population, methods to attribute selection and levels, approach to experimental study design, types of statistical model, number of attributes, participants, disease category, and preferences.

## Synthesis of results

We used DSD model to synthesis the extracted data. DSD model accommodate client's unique preference that has been utilized to retain clients in long-term HIV/AIDS care [21] and recommended to be applied in chronic diseases management [22]. Although there is limited evidence of the DSD model being explicitly named and applied by listing its domains, healthcare services have been provided through various health professionals across different settings based on client preferences. Several programs for NCD care align with DSD model. For instance, the Integrated Virtual Diabetes Care Clinics in Western Sydney, Australia, offer different service options based on patient complexity, using a combination of virtual and in-person consultations to tailor care according to patient needs and risk levels [23]. Key elements of a DSD for NCD include service frequency (when), service location (where), health worker (who), and service packages (what) [24,25]. We grouped similar findings based on the words they reported into the closer category. With adapting the DSD model, the

main groupings were 'when' regards to follow-up time, frequency of visit, date and time of services; 'where' to present the service location, such as clinics, health center, hospitals, schools, or community residents; 'who' to see who are preferred by clients whether nurses, general practices, specialists, or multidiscipline; and 'what' domain represent the services package, for which we replaced 'what' with 'how' in this review. To support the synthesis of the main findings, particularly to the how domain, we used WHO's PHC framework. This framework grasped the quality of care components: timely access, efficiency, safety, effectiveness, and other PHC functions [26].

## Results

### Search results

A total of 33 articles were included in the current review after screening of 3,185 articles (Figure 1).

Two studies focused on general NCD [27,28]. Twenty-four studies were on cancer: non-specific types of cancer [29–34]; breast cancer [35–39]; skin cancer (melanoma) [40–42]; prostate cancer [43–45]; colorectal cancer [46,47]; cancer or blood disorders [48]; multiple myeloma [49]; lung cancer [50]; papillary thyroid cancer [51]; and oesophagogastric, bowel, or lung cancer [52]. Three studies were on DM: type 1 DM [53] and type 2 DM [54,55]. Three studies addressed mental health: mental health prevention [56] and depression and anxiety [57,58]. One study focused on dementia, heart failure, and cancer [59]. When we see the immigration status of the study population, only one study conducted among Indian immigrants living in Australia [54], while others did not mention immigration status. Supplementary file 2 presents the details of included articles (S2 Table).

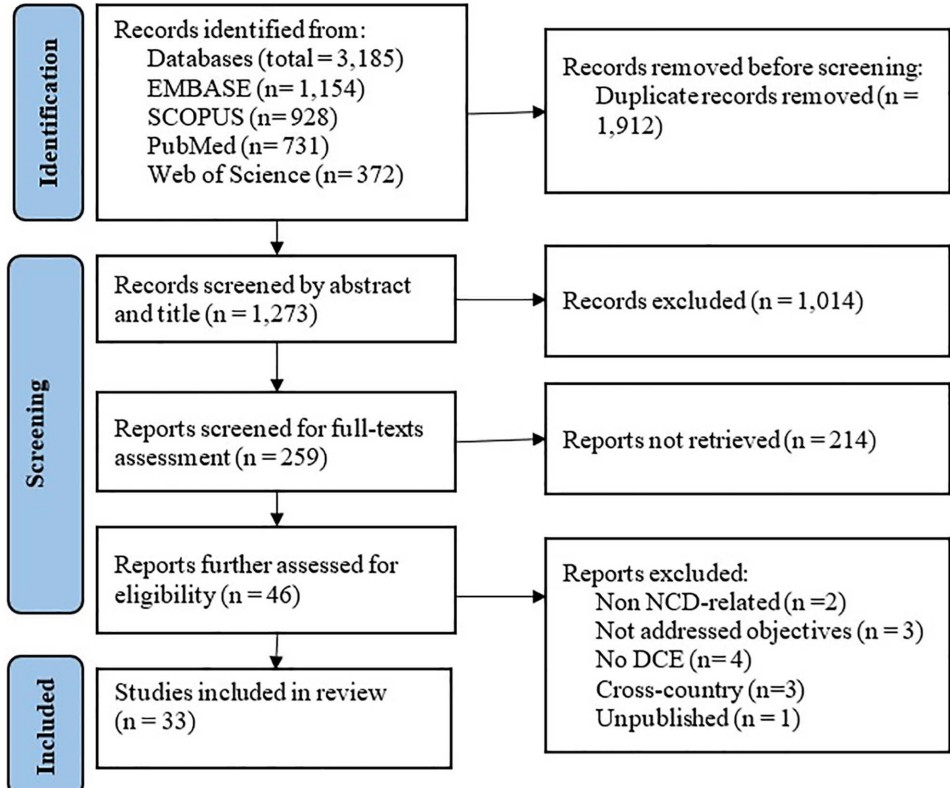

**Fig 1. PRISMA Flow Chart for articles selection process.**

The preferences of attributes include the following services: parenting interventions for mental health prevention [56], support for children with diabetes at the school for mental health [53], diagnostic tests and screening services [27,31,33,41,42,44,47,50,52,58], medication treatments [39,40,54,55], radiation therapy [36,43], surgical management [37,46,51], pharmacy services [28], follow-up services [35,38], emotional or other support services [30,48], treatment decision [29], care [57], end of life care [34,59], and appointment services [32]. To summarize the available articles according to the continuum of care and specific activities: one study each focused on health promotion (communication) and prevention (communication), ten on screening (check-ups), and twenty on treatment (seventeen care and three on continuity of care) (Table 1).

## Preferences of Individuals with NCD

Clients attending NCD services preferred essential attributes, and most are related to quality-of-care elements. The preferred model of care needs to be cost effective, accessible both in distance and time, less frequent follow-ups during no need for frequent follow-up, and care by better-trained care providers. Clients attending attributes and preferences are framed in who, how, where, and when dimensions (Table 2).

**Who:** Overall, clients preferred well-trained and educated care providers, with few specific distinctions. They preferred health workers over themselves in a sample collection [41,42], specialists over general practitioner (GP) or a dermatology-specialised GP for dermatologist care [41,42]. They preferred screen by a cancer nurse and follow-up care by mental health professionals embedded within the cancer care team for anxiety and depression in cancer clinic [58], breast physician followed by breast care nurse [35], medicines supply by a pharmacist for regular and symptom flare-up medicines [28], emotional support either by counsellors (psychologist) and/or peers over none [48], clinician-directed referral for great concern of depression over self-directed approach [57], having a school counsellors over not having them [53], and a surgeon with additional training [46]. Clients preferred primary care physician over a genetic specialist for a polygenic test for cancer [31].

**How:** This element includes cost, effectiveness, accessibility, timeliness, efficiency, comprehensiveness, continuity of care, and patient-centeredness. To illustrate, affordability include lower cost [27,31,41,42,44,47,50,52,56,58], funding for treatment [50], did not impact life insurance eligibility or premiums [31]; effectiveness involves diagnosis and testing accuracy [27,31,41,42,44,47], effective outcomes [44,50], safe procedures and/or lower side effects [31,41,42,44,47,50]; accessibility and timeliness cover short waiting time to access care [27,41,42,50,52,58], wait time to get result [41,42,50,52], screening process time taken [58], time away from usual activities including travel (shorter travel time) [41,42,52]; efficiency refers to efficient services in terms of time [56]; comprehensiveness includes tested for multiple cancer type [31]; continuity of care involves GP familiarity [52]; and patient-centeredness includes clients participating in decision making [46] and privacy [31].

**Where:** Clients preferred face-to-face for anxiety and depression screening [58] and a polygenic test for cancer [31], follow-up service for breast cancer survivors (followed by alternate face-to-face and telephone) [35]. They preferred care local breast cancer clinic follow-up service for breast cancer survivors [35] and a pharmacy located within a 'one-stop' health center followed by home delivery of medicines [28], while they in contrast preferred treatment in a teaching hospital over district hospital for surgical management for colorectal cancer [46] and a cancer center over a general hospital setting [32]. Test samples processed and analyzed in Australia were preferred over those processed overseas for lung cancer [50].

**When:** Clients generally preferred less frequent follow-up appointments [35,51,55]. For example, follow-up services every 6 months for breast cancer survivors [35], and injection frequency for DM (once weekly over twice weekly) [55]. Screening for anxiety and depression in cancer care with a regular monthly or every 3 months interval over one year [58].

## Social determinants of health on clients preferences

Social determinants of health influence client's preference across the continuum of NCD care. These determinants include place of resident, occupation, gender, education status, socioeconomic status, and age. Rural participants preferred

**Table 1. Authors, study population, disease category, and services.**

| Authors | Study population | Disease category | Service studied | Continuum of care |
|---|---|---|---|---|
| Ahmed A et al 2021 [54] | 18 years or older born in India and live in Australia | Type 2 DM[1] | Conventional vs. Ayurvedic medicines | Treatment (Care) |
| Bessen T et al 2014 [35] | Breast cancer survivors | Breast cancer | Follow-up services in the absence of long-term specialist-based care | Treatment (Continuity of Care) |
| Broomfield G et al 2022 [56] | 18 years or older parents of birth to 18 years child | Mental health prevention | Internet- and mobile-based parenting interventions | Prevention (communication) |
| Brown A et al 2022 [43] | 18 years and over men with and without prostate cancer | Prostate cancer | Image-guidance in prostate radiation therapy | Treatment (Care) |
| Fifer S et al 2018 [55] | 18 years or older with T2DM and on either injectable or oral medicines | Type 2 DM | Treatments | Treatment (Care) |
| Fifer SJ et al 2020 [49] | Patient with multiple myeloma and, carer, physician, and nurse preferences working on multiple myeloma | Multiple myeloma | Treatments | Treatment (Care) |
| Fifer S et al 2022 [50] | 18 years or older with stage III or IV Non-Small Cell Lung Cancer, and Clinicians who treated a minimum of five patients with this cancer | Non-small cell lung cancer | Genetic and Genomic Testing | Screening (Check-up) |
| Goodall S et al 2012 [48] | 16 to 32 years old, with cancer or a blood disorder, and carers | Cancer or a blood disorder | Support services | Treatment (Care) |
| Herrmann A et al 2018 [29] | 18 years or over with cancer and presented for their second or subsequent outpatient oncology consultation | Cancer | When and how to make treatment decision | Treatment (Care) |
| Hobden B et al 2018 [57] | 18 years or older oncology patients | Depression | Care | Treatment (Care) |
| Howard K et al 2014 [44] | 40 to 69 years old men who diagnosed or treated for prostate cancer | Prostate cancer | Screening | Screening (Check-up) |
| Howard K et al 2023 [36] | Women aged 40–79 years with early-stage breast cancer and from the general population | Breast cancer | Radiation Therapy | Treatment (Care) |
| Kenny P et al 2024 [59] | 45 years or older adults from the general population | Cancer, dementia, and heart failure | Care at the End-of-Life Care | Treatment (Care) |
| Livingstone A et al 2023 [40] | Adults with resected stage III melanoma | Melanoma | Adjuvant Immunotherapy | Treatment (Care) |
| De Abreu Lourenço R et al 2019 [37] | A general community sample of 18 years or over women | Breast cancer | Contralateral prophylactic mastectomy | Treatment (Care) |
| Nickel B et al 2018 [51] | 18 years or over without previously diagnosed or treated for thyroid cancer | Papillary thyroid cancer | Treatment (full surgery, partial surgery, and monitoring) | Treatment (Care) |
| Salkeld G et al 2005 [46] | 18 years and over | Colorectal cancer | Surgical management | Treatment (Care) |
| Senanayake S et al 2024 [38] | Breast cancer survivors completed treatment within the last five year | Breast cancer | Follow-up care | Treatment (Continuity of care) |
| Snoswell CL et al 2018 [41] | Adults with not a melanoma within the last five years and had access to Mobile Phone compatible with the dermoscopic attachments | Skin cancer | Screening (mobile teledermoscopy) | Screening (Check-up) |
| Spinks J et al 2016 [42] | 50 to 64 years with moderate or high risk for melanoma | Melanoma | Screening (mobile teledermoscopy) | Screening (Check-up) |
| Turon H et al 2020 [30] | 18 years or older adults with cancer | Cancer | Information provision about emotional concerns during procedure for cancer | Treatment (Care) |
| Venning B et al 2022 [31] | General population | Cancer | A polygenic test | Screening (Check-up) |
| Whitty JA et al 2015 [28] | Clients with chronic conditions and health workers | Chronic conditions | Pharmacy service for chronic conditions | Treatment (care) |

*(Continued)*

**Table 1.** (Continued)

| Authors | Study population | Disease category | Service studied | Continuum of care |
|---|---|---|---|---|
| Wong SF et al 2016 [32] | Adults with cancer | Cancer | Appointment services | Treatment (Continuity of care) |
| Yim J et al 2021 [58] | 18 years or older with history of cancer | Anxiety and depression | Screening | Screening (Check-up |
| Yu A et al 2021 [33] | Adults with cancer who received chemotherapy | Cancer | Assessment of peripheral neuropathy due to chemotherapy | Screening (Check-up) |
| Howard K et al 2011 [47] | Patients with clinical indications suspicious of colorectal cancer | Colorectal cancer | Computed tomography colonography | Screening (Check-up) |
| Liede A et al 2017 [39] | Women aged 25–55 years | Breast cancer | Breast cancer risk reduction hypothetical drug | Treatment (care) |
| Ride J et al 2024 [53] | School-aged children with type 1 DM | Type1 DM | Support for children with diabetes at the school for mental health | Health promotion (communication) |
| Senanayake S et al 2024 [27] | General population aged 18 years or older | NCD[2] | Screening service | Screening (Check-up) |
| Venning B et al 2024 [52] | 40 years or older of public | Oesophagogastric, bowel, or lung cancer | Testing for symptoms | Screening (Check-up) |
| Waller A et al 2018 [34] | Patients and support persons | Cancer | End of life care | Treatment (care) |
| Sigurdson S et al [45] | Patients with prostate cancer receiving local external beam radiation therapy | Prostate Cancer | Hypofractionated Radiation Therapy | Treatment(care) |

[1]Diabetes mellites; [2]noncommunicable diseases

shorter external beam radiation therapy and were less willing to extend treatment to reduce toxicity risk [45]. Individuals employed fulltime were less likely to prefer full surgery than not employed fulltime [51]. Differences in preferences between males and females were reported. In cancer-related care and support, males valued assistance with reintegration into work or study, emotional support for their families, and less emotional support for themselves. They placed lower value on culturally specific support compared to females [48]. Males favoured specialist care and preferred to be accompanied by family or friends during cancer treatment [32]. They were also more likely to choose a self-directed approach and least preferred clinician-directed referrals, whereas the opposite was true for females [57]. Females showed a lower preference for melanoma immunotherapy than males [40].

Education level and socioeconomic status influence clients' care preferences. Parents without formal postsecondary education favoured briefer modules and a predefined module order (less user-controlled) [56]. Those who completed high school or higher were more likely to prefer computed tomography colonography (CTC) [47]. Individuals with a tertiary education preferred more invasive forms of testing for cancers-related symptoms [52]. However, education level did not affect preferences for prostate-specific antigen screening [44]. Health literacy also influenced preferences: those with lower health literacy were less likely to prefer less invasive treatment options [51]. Regarding socioeconomic status, lower-income parents preferred app-based, inexpensive, and shorter programs in health education [56]. In contrast, income level did not affect preferences for prostate-specific antigen screening [44], and having private health insurance did not significantly influence test preferences [47].

Age of clients was crucial to in informing care models. Increasing age was associated with a lower preference for prostate-specific antigen screening compared to no screening [44] and a lower preference for CTC [47]. Age did not influence oncology patients' preference for depression care [57]. Another study addressed differences in surgical preferences

**Table 2. Continuum of care, services, and preferences for NCD.**

| Continuum of care | Services | Preferences (Who, Where, When, How) |
|---|---|---|
| Health promotion | • Technology-enabled parenting for mental health prevention [56]<br>• Psychosocial support at school | **How:** cost: cheaper programs [56] and efficiency: briefer modules [56]; having a school counsellor [53]; support to attend off-campus or on-campus activities [53], coping skills training [53] |
| Screening and diagnosis | • Assessment of nerve damage associated with chemotherapy treatment [33]<br>• Genomic profiling in non-small cell lung cancer [50]<br>• Prostate cancer screening [44]<br>• Screening via mobile Teledermoscopy [41,42]<br>• A polygenic test for cancer [31]<br>• Screening for mental illness in cancer clinic [58]<br>• Computed tomography colonography and colonoscopy as diagnostic tests [47]<br>• Screening service for NCD [27]<br>• Diagnostic test for oesophagogastric, bowel, or lung Cancer [52] | **Who:** Healthcare worker choices over self-examination [41,42], a dermatologist over a standard general practitioner (GP) or a dermatology-specialised GP [41], dermatologists over GP [42], primary care physician over a generic specialist [31], screen by a cancer nurse and follow-up care by mental health professionals embedded within the cancer care team [58], and physician test (e.g., done by physician or client activity) [33]<br>**Where:** face-to-face [58], test sample processed and analyzed in Australia over sample processed in overseas [50], primary care physician over online or genetic specialist [31]<br>**When:** regular interval (monthly or every 3 months over one year) [58]<br>**How:** accessibility: short waiting time to access care [41,42,50,58]; time away from usual activities including travel [41,42,52], wait time to screening appointment [27], wait time to get result [41,42,50], screening process time taken [58], waiting time to be screened [58], test and result waiting times [52]; information about the importance of services [27]; affordability [27,31,41,42,44,47,50,52,58], preferring lower cost [41,50,58], funding for treatment [50], no effect on life insurance eligibility or premiums [31]; effectiveness: diagnosis and testing accuracy [27,31,41,42,47], prostate cancer diagnoses [44]; comprehensiveness: tested for multiple cancer type [31], enabled cancer risk reduction [31]; reporting: chance the result will change cancer screening [31]; interpretation and reporting [50], screening conduct [27], cancer type [31], assessment impact on clinical time [33], assessment results on influence care/treatment [33]; patient-centeredness: assessment on symptoms or impacts due to symptoms [33], level of assessment (detailed or major nerves) [33], questionnaire for assessment [33], privacy [31]; efficiency: testing process [31], the testing strategy [52]; continuity of care: GP familiarity [52]; equity: Screening service routine or based on request [58]; health benefits: potential mortality benefit [44], prostate cancer deaths [44], chance of actionable outcome [50]; safety and side effects: unnecessary biopsies [44], risk reduction measures [31], negated the need for additional biopsy [50], test characteristics [44], bowel preparation [47], needing a second therapeutic procedure after CTC [47], impotence [44], urinary incontinence/bowel problems [44], a decrease in benign mole removals for each diagnosed case of skin cancer [42], number of genes tested [50], excision ratio for skin cancer detection [41,42], method of screening [41,42], tissue requirements [50], germline findings [50] |

*(Continued)*

| Continuum of care | Services | Preferences (Who, Where, When, How) |
|---|---|---|
| Treatment | • Medications: Conventional vs. ayurvedic medicines for type 2DM [54]; Injectable treatment for Type 2 DM [55]; Treatment for multiple Myeloma [49]; Adjuvant immunotherapy for resected stage III melanoma [40]; Breast cancer risk reduction hypothetical drug [39] | **How**: neither were preferred with a greater non-preference for ayurvedic medicine [54]; route of drug administration: greater preference for oral treatments [49]; side-effects: lower risk of mild side effects [49], no severe side effects [40,49,55]; health effects: weight change [55], longer periods of overall survival [49], lower probability of recurrence [40], and longer remission periods [49]; affordability: lower out of pocket costs [40,49], |
| | | **When**: Injection frequency (once weekly over twice weekly) [55] |
| | • Survivorship follow-up services in breast cancer care [35,38]<br>• Supportive services for cancer or blood disorder [48]<br>• Consultation for treatment decision [29]<br>• Depression care [57]<br>• Information provision about emotional concerns for adults with cancer [30]<br>• Appointment services for cancer [32]<br>• Support for children with diabetes at the school [53]<br>• Pharmacy services [28]<br>• Contralateral prophylactic mastectomy for breast cancer [37]<br>• Treatment preference [51]<br>• Surgical management for colorectal cancer [46]<br>• Image-guidance in prostate radiation therapy [43],<br>• Hypofractionated Radiation Therapy for Early-Stage Breast Cancer [36] and [45] | **Who**: Breast physician followed by Breast Nurse [35] and medical specialist in a cancer center [32], medicines supply by a pharmacist [28], emotional support either by counsellors and/or peers [48], clinician-directed referral for great concern of depression [57], self-directed approach for male [57], having a school counsellor [53], specialty training of the healthcare provider shaped patients' treatment choices [46], who decides treatment [46]<br><br>**When**: Follow-up every 6 months [35], less frequent follow-up [51], participants most preferred to receive the information 1 week before the procedure, followed by 3 days prior, and finally on the day of the procedure [30]<br><br>**Where**: Local breast cancer clinic [35], drop-in clinics followed by secondary prevention [35], face-to-face attendance followed by a combination of face-to-face and telephone) [35], a 'one-stop' health center followed by home delivery of medicines [28]; type of hospital [46]<br><br>**How**: continuity of care: seen by usual doctor [32], surgeon's communication [46]; accessibility: shorter appointment duration to see a doctor [32], travel shorter [32]; timeliness: two shorter consultations rather than one longer consultation [29], shorter treatment duration [36]; support: family/friends stay with clients overnight [32], emotional support for their family [48], financial support and assistance returning to school/work over services relating to cultural and spiritual needs [48], avoiding relocation [36]; affordability: lower cost for services [32,43,51]; communication: face-to-face discussion and written materials over website [30]; person-centredness: participants not choice new services if there is a high quality services [28]; effectiveness: accuracy [43]; influence on daily activities and lower side effects [36,43,51] |
| Rehabilitation | • End of life care for cancer [34]<br>• Care for older people with cancer, dementia, and heart failure at the End of Life [59] | **How**: severity of the problem: effect of life extension (not associated) [34], increased consciousness [34], decreased pain preferred [34], patient symptoms [59], informal carer stress [59]; affordability: lower cost for services [59] |

between younger and older women in breast-cancer. Younger women preferred breast-conserving surgery, while older women were less influenced by the type of surgery chosen [36]. Age and quality of life also influenced care preferences. Younger women with higher quality of life favoured multidisciplinary care teams (including specialists, nurses, and general practitioners) and valued shared survivorship care plans. In contrary, older women with lower quality of life were more concerned about out-of-pocket costs and remained neutral about team composition [38]. Older preferred shorter external beam radiation therapy and were less willing-to-extend treatment to reduce toxicity risk [45]. Those aged ≥60 years preferred their regular GP and more invasive forms of testing for cancer-related symptoms [52].

## Discussion

This review provides the preferences of clients with NCD. The review identified that the ideal model of care included a well-trained provider, affordability, accessibility in terms of both distance and time, person-centredness, safety, efficiency, effectiveness, comprehensiveness, relational continuity, and less frequent follow-ups, except for frequent screening follow-ups for mental health disorders.

Well-trained health worker was preferred to deliver NCD services. Clients prefer highly trained providers because they feel more confident and trust the care they receive from well-trained and educated health care providers [60]. Clients also preferred clinician-directed referral for great concern of depression over a self-directed approach [57]. This aligns with the existing routine care standards. For instance, in Australia, mental health issue referral occurs after the client consults a GP or telephone triage services [61]. However, health workers may not necessarily fully engage in some NCD-related services. For instance, taking sample from the cervix is advisable to be done by clients because self-sampling is cost effective and user friendly [62] which is highly preferred and recommended in high-income countries, including Australia [63,64]. Since health workers are the backbone of the health system [65], most care models include a multidisciplinary or interdisciplinary team [13]. With the inclusion of health workers, continuous professional development needs to be part of the care model. There is a good example that the Australian Aged Care Quality and Safety Commission has introduced that providers need to have a written plan for continuous improvement using 'Plan-Do-Check-Act' model [66].

Care models should satisfy clients need for essential elements of quality of care. According to the current review, clients preferred accessible, person-centred, safe, efficient, effective, and affordable care. This review presented the clients preferences in the how dimension of DSD that are in line with some of the principles of the primary care model: accessibility, continuity, comprehensiveness, and patient-centred [67] and other quality of care elements: equity, safety, effectiveness, timeliness, and efficiency [68]. Clients in other countries also exhibited similar preferences [69–71]. Since clients prefer accessible services, there are specific health care models in Australian health system, aimed at availing services closer to the community. For instance, eligible Australians aged 45–74 and higher risk individuals can receive a free kit to collect stool samples at home to check for traces of blood that could indicate bowel cancer; this test is mailed directly to individual's home every 2 years after the last screening test is completed [72]. Clients' preferences are considered in this service comprising lower cost for receiving free kits, accessible with no travel distance and waiting time, safety with sample taken from the stool, and follow-up with less frequent sample collection and testing. However, the community members' preferences for the attributes of this model of care were not fully examined, although modifiable features of home bowel cancer screening kits were identified. Participant preferences were evaluated in terms of how bowel cancer screening kits can be modified [73,74]. Users favoured collection devices designed to reduce exposure to faecal material, with smaller packaging, easy-to-follow steps, and illustrated guidelines [74]. Clients also preferred a care model that engaged them in the decision-making process, considering their privacy, familiarity and communication with health workers, reflecting that person-centeredness and continuity of care in clients' interest. This might be due to communication strategies positively impacted patient-centred outcomes [75]. Affordable care was one of the attributes of service provision. For instance, study respondents indicated a stronger preference for monetary support and assistance with returning to education or employment rather than for cultural or spiritual services [48]. Although Medicare covers, many healthcare needs in Australia, the

costs of accessing healthcare services may remain a concern among some groups of individuals [76]. Between 2006 and 2014, nationally representative data revealed that individuals within the lowest income group (first decile) were more likely to incur catastrophic health costs compared to those in the top income group [77].

The preferred locations or modalities for NCD services in Australia were variable due to the specificity of the services. Clients preferred face-to-face for anxiety and depression screening [58], a polygenic test for cancer due to patients may have been influenced by the vignette [31], and follow-up service for breast cancer survivors [35]. This might be due to telehealth was considered resource-intensive and increased remote clinic staff workload, as staff often had to assist patients during sessions, handle administrative tasks, and provide interpreter services [78]. Health workers replied that telehealth cannot replace in-person care [78]. Clients did not prefer telehealth to face-to-face when physical examination was necessary, although consumers value telehealth availability and appropriate use [79], and women would be prepared to accept alternating method (a sequential hybrid care pathway: face-to-face followed by telephone) if the frequency of contact with the follow-up service is maintained [35]. It is assumed that telehealth was most effective when patients had an existing relationship with their provider, understood their health well, spoke English, and were comfortable using digital technology in Australia [78]. Another study among clients with chronic musculoskeletal conditions revealed that 43% of participants would prefer home telehealth over having to travel to attend their appointments [80]. Similarly, one-third of general clients preferred a telehealth over in-person visit in the United States of America, particularly female preferred it more likely than male [81]. The mHealth app, comprising a smartphone interface, clinician portal, and secure cloud data, was well received by women with gestational diabetes and clinicians [82]. Home telemonitoring for chronic diseases management requires empowering clients, maintaining client-clinician interactions, assigning a dedicated telemonitoring clinical care coordinator, translating telemonitoring services into clinical pathways, and engaging health care teams [83]. This implies that mHealth care and face-to-face service preferences may depend on the health disorder conditions, the need for physical examinations, other conditions that require physical presence in a health care setting, and social determinants of health. For example, older adults faced technical barrier, including digital skills in telemedicine care model, according to another review [84]. The National Survey of older US adults on willingness to use telehealth revealed that willingness to use telehealth declines with increasing age, but a lower cost or an insurance coverage and perceived usefulness modify their willingness to use it [85]. Older (≥80 years) rural patients with chronic conditions in New South Wales preferred telephone over online mediums (Skype or Zoom) [86]. Australian Communication and Media Authority highlighted the importance of supporting older people's digital literacy and providing them with the skills to navigate what can be confusing and potentially risky environments to help them use the internet and engage with the digital world [87].

A specialist multidisciplinary wound clinic and telehealth services (real-time video consultation for remote settings) increased quality of life and access to evidence-based practices for clients eligible for Bulk Bill in Queensland [88]. Care team consultations, through in-person visits and phone calls, are supported by an easy-to-use online platform that provides access and reminders for upcoming GP appointments and prescriptions [89].

The client's preference on service provision date and time was not included as attributes in the studies included in the current review. In other low- and middle-income countries, clients with NCD preferred the same day time of testing and treatment with diagnosis [51] and different access times [5]. In the current review, clients preferred less frequent follow-up appointment services, with few exceptions. For example, follow-up services every 6 months for breast cancer survivors [35], and injection frequency for DM (once weekly over twice weekly) [55]. The exception was screening for anxiety and depression with a regular monthly or every 3-month interval over one year [58].

Furthermore, social determinants of health shape how clients with NCD influence client's preference across the continuum of NCD care. Clients with different sociodemographic status had different preference for different care contexts due to the accessibility, equity, person-centredness, comprehensiveness, continuity of care, effectiveness, cost-effectiveness and quality of care. For example, rural residents preferred shorter treatment schedules due to travel related cost, in the current review, explained by participants from rural participants preferred shorter therapy and may also take risk of side

effects stream from the treatment due to being shorten [45]. Cost-effectiveness was also essential and linked with age, like older women with lower quality of life valued out-of-pocket costs [38]. Likewise, lower-income parents preferred app-based, inexpensive, and shorter programs in health education [56]. Preferences are varied due to unique needs or preexisting demands, value and social contexts, such as males valued family involvement and work reintegration, and females value cultural and emotional specific support [48]. Person-centredness could be demonstrated in the development of a model of care with consideration to clients' age, as body image and self-esteem may be linked to age. For example, younger women often prefer breast-conserving surgery [36]. Comprehensiveness of care due to involvement of multidisciplinary team, particularly, preferred by younger women with higher quality of life [38]. Older preferred their regular GP that is continuity of care difference [52]. In conducting this review, the search for articles was limited only to those published in English. One reviewer (AE) conducted the article screening process, though the team provided input at each stage. This review is also entirely based on articles using DCE.

## Conclusions

This review highlights that clients prefer NCD care models that are accessible, person-centered, safe, efficient, and effective. Well-trained healthcare providers, self-directed approach (for men) or clinician-directed referrals (for women), and multidisciplinary teams (e.g., preferred by younger women with higher quality of life) are key components of preferred care models. While telehealth and mHealth services are valued for convenience, face-to-face interactions remain essential for conditions requiring physical assessments. Home-based services, such as self-sampling for cervical cancer and mailed bowel cancer test kits, are appreciated for their accessibility and ease of use. This implies that in informing the development of models of care, clients' individual characteristics may influence the selection of care attributes or elements, as social determinants of health can be linked to aspects such as accessibility, equity, comprehensiveness, continuity of care, effectiveness, cost-effectiveness, and overall quality of care. In addition, the severity of diseases condition, nature of therapy, and social and cultural values are crucial in the development of person-centred model of care. The review reinforces the importance of strengthening PHC-based health systems to meet client preferences and improve NCD management.

## Supporting information

**S1 PRISMA Checklist.** PRISMA Checklist.
(DOCX)

**S1 Table.** Search strategies.
(DOCX)

**S2 Table.** Characteristics of included articles.
(DOCX)

## Acknowledgments

**Authorship Confirmation**: All authors certify that they meet the ICMJE criteria for authorship.

## Author contributions

**Conceptualization:** Aklilu Endalamaw, Yibeltal Assefa.

**Data curation:** Aklilu Endalamaw, Darsy Darssan, Resham B Khatri, Yibeltal Assefa.

**Formal analysis:** Aklilu Endalamaw.

**Investigation:** Aklilu Endalamaw.

**Methodology:** Aklilu Endalamaw.

**Project administration:** Aklilu Endalamaw, Yibeltal Assefa.

**Supervision:** Yibeltal Assefa.

**Validation:** Aklilu Endalamaw, Darsy Darssan, Resham B Khatri, Yibeltal Assefa.

**Visualization:** Aklilu Endalamaw, Darsy Darssan, Resham B Khatri, Yibeltal Assefa.

**Writing – original draft:** Aklilu Endalamaw.

**Writing – review & editing:** Aklilu Endalamaw, Darsy Darssan, Resham B Khatri, Yibeltal Assefa.

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
