## [Decision Letter · Decision Letter 0]

9 Oct 2025

PGPH-D-25-01773

Client Preferences in Noncommunicable Diseases Management in Australia: A Scoping Review

Dear Dr. Endalamaw,

Thank you for submitting your manuscript to PLOS Global Public Health. After careful consideration, we feel that it has merit but does not fully meet PLOS Global Public Health’s publication criteria as it currently stands. Therefore, we invite you to submit a revised version of the manuscript that addresses the points raised during the review process.

We look forward to receiving your revised manuscript.

Kind regards,

Dr Buna Bhandari

Academic Editor

Journal Requirements:

2. We have noticed that you have uploaded Supporting Information files, but you have not included a list of legends. Please add a full list of legends for your Supporting Information files after the references list.

Additional Editor Comments (if provided):

Reviewers' comments:

Reviewer's Responses to Questions

**Comments to the Author**

1. Does this manuscript meet PLOS Global Public Health’s publication criteria?

Reviewer #1: Yes

Reviewer #2: Partly

2. Has the statistical analysis been performed appropriately and rigorously?

Reviewer #1: Yes

Reviewer #2: Yes

3. Have the authors made all data underlying the findings in their manuscript fully available (please refer to the Data Availability Statement at the start of the manuscript PDF file)?

Reviewer #1: Yes

Reviewer #2: Yes

4. Is the manuscript presented in an intelligible fashion and written in standard English?

Reviewer #1: Yes

Reviewer #2: Yes

Reviewer #1: The article did not describe the ages of those who preferred in person visits. It did not differentiate between older and younger populations and their preference for telemedicine. Did the older patients with NCDs have access to phones and know how to do telemedicine or was it a select group that did telemedicine amongst the NCD population?

Reviewer #2: This is an important article that engages with the suitability of models of primary healthcare for specific NCD management and identifies areas for improvement within the Australian context. While this is based on the synthesis of papers on chronic conditions, it could benefit from further clarification.

All the included papers were written during the second decade of the 21st century. This paper, though, mentions chronic care models in Australia. However, it could highlight the kinds of DSD programmes available for the NCDs to give a backdrop to the NCD management.

Page 4 Under Eligibility Criteria

“We included preference articles used discrete choice experiment (DCE) among clients with major NCD in general or specifically on cardiovascular diseases (CVD), cancer, chronic obstructive pulmonary diseases (COPD), diabetes, or mental health disorders”. – Out of 32 included papers, the authors can specify how many are under each NCD marked here?

The analysis can benefit from additional inputs and enhance its richness. In the analysis section, disaggregating the 'client' can be helpful in understanding how NCD services are experienced across different demographics, including sex and gender, ethnicity, nationality, and region.

The paper draws upon the DSD model. In conclusion, while there is generalisation, it may be strengthened by adding some of the management-specificities of the specific NCD, which can be further addressed in the NCD management. Its implications for integrating healthcare at the PHC level can be highlighted.

Table 1 can be reorganised NCD wise

In Figure 1 the box for Screening and Incuded overlap. This may be checked.

**Do you want your identity to be public for this peer review?** For information about this choice, including consent withdrawal, please see our Privacy Policy

Reviewer #1: No

Reviewer #2: No

---

## [Editor Report · Decision Letter 1]

16 Nov 2025

Client Preferences in Noncommunicable Diseases Management in Australia: A Scoping Review

PGPH-D-25-01773R1

Dear Mr. Endalamaw,

We are pleased to inform you that your manuscript 'Client Preferences in Noncommunicable Diseases Management in Australia: A Scoping Review' has been provisionally accepted for publication in PLOS Global Public Health.

Best regards,

Dr Buna Bhandari

Academic Editor